# Reliability of a Custom Device Used to Measure Isometric Knee Flexor and Extensor Strength in Standing Position

**DOI:** 10.3390/life13020458

**Published:** 2023-02-06

**Authors:** Tommaso Minuti, Paolo Cigni, Michele Costagli, Alessandro Cucini, Erika Cione, Samuele Melotto, Stefano Rapetti, Leonardo Ricotti, Roberto Cannataro

**Affiliations:** 1The BioRobotics Institute, Scuola Superiore Sant’Anna, 56127 Pisa, Italy; 2Department of Excellence in Robotics & AI, Scuola Superiore Sant’Anna, 56127 Pisa, Italy; 3Auxilium Vitae Volterra Spa, Borgo San Lazzaro, 5, 56048 Volterra, Italy; 4Italian Society of Nutrition, Sports and Well-Being (SINSeB), Via Luigi Cherubini, 2, 50053 Empoli, Italy; 5Fisioclinic Dott. Paolo Cigni, Via Porta Massetana 1, 56045 Pomarance, Italy; 6Department of Pharmacy, Health and Nutritional Sciences, University of Calabria, 87036 Rende, Italy; 7Galascreen Laboratories, University of Calabria, 87036 Rende, Italy; 8Associazione Calcio Milan, Via Aldo Rossi 8, 20149 Milano, Italy; 9Research Division, Dynamical Business & Science Society—DBSS International SAS, Bogotá 110311, Colombia

**Keywords:** muscle strength assessment, dynamometer, knee flexors, extensor, maximum force, isometric strength

## Abstract

*Background:* Assessing lower limb strength in the field is problematic, as the “gold standard assessment” with isokinetic strength is cumbersome, and the device is costly and not transportable and keeps the angle of the hip at around 90°. *Methods*: We evaluated isometric muscle strength in a standing position with the help of an exoskeleton that holds the subject and makes the test easily repeatable. *Results:* The optimal device angles for hip and knee were, respectively, 20° and 80° for flexor tests and 30° and 40° for extensor tests. Test–retest reliability was very high for the right knee extensor (ICC 0.96–0.98), left knee extensor (ICC 0.96–0.97), right knee flexor (ICC 0.91–0.96), and left knee flexor (ICC 0.96–0.97). Furthermore, the typical error in percent (T.E.%) ranged from 2.50 to 5.50%, and the change in the mean in percent ranged from 0.84 to 7.72%, making it possible to determine even a slight variation in force. *Conclusions:* this new method could represent a valid alternative for assessing strength, due to the high reliability and the favorable joint position, particularly in football.

## 1. Introduction

Muscle strength is responsible for joint torque production. Therefore, it is crucial in determining the efficacy of human body movements and guaranteeing joint stability and posture maintenance [1]. Muscle strength has been the object of many studies in the sports field, in which the relationship between muscle strength, training, level of competition, and performance has been explored [2,3,4]. Moreover, strength training is increasingly being used in managing various conditions such as sarcopenia [5] or osteoarthritis [6].

Therefore, muscle strength assessment provides to medical staff, coaches, and athletic trainers, at any level of competition, information about muscle weakness and imbalances in order to (1) monitor the athletic condition of a player over time, (2) guide the recovery process from an injury, or (3) identify specific deficits in strength potentially increasing injury risks early, and correcting them with ad hoc training.

The role of muscle strength as a possible risk factor for sports injuries is hotly debated due to the complexity and multifactoriality of the event. However, among team sports, soccer is the most prone to produce injuries [7]. According to a study promoted by UEFA, 87% of soccer injuries involve lower limbs, of which 17% concern muscle injuries (12% flexors and 5% extensors) and 5% traumas to the anterior cruciate ligament (ACL) [8].

Recent studies showed that hamstring and quadricep weakness and imbalances did not correlate with injuries [9,10,11]. Nevertheless, Thorlund et al. demonstrated that a lower hamstring-to-quadricep muscle activation level was correlated with ACL injury in handball and soccer players [12]. Furthermore, Croisier et al. found that restoring the balance between agonist and antagonist muscle groups considerably decreased the risk of injuries [13]. Such a lack of consensus in the scientific community raises the need to explore this topic further.

In this regard, assessing the muscle strength of the lower limbs has evolved significantly in recent years. As a result, state-of-the-art methods have been proposed to assist athlete musculoskeletal profiling, monitoring, and training dose control.

Isokinetic dynamometry is considered the “gold standard” screening tool for assessing hamstring and quadriceps [14]. However, its use is limited due to the high cost of the device, lack of portability, and time needed to complete the assessment, especially when a large number of athletes are tested in succession [15]. Therefore, research has proposed alternative methods for measuring knee flexor and extensor strength, such as the Nordic hamstring test to assess eccentric bilateral strength or the vertical jump test to evaluate explosive strength [16]. Although these screening tools are reliable and customarily used, there is an increased risk of injury with eccentric load produced during the execution of the test [17,18]. A possible alternative that overcomes the limitations associated with dynamic force assessment is the use of isometric tests [17,18,19].

A manual muscle strength test (MMT) has historically been the most common method for assessing muscle strength. MMT is a subjective technique that uses a 5-point grading scale and requires no testing equipment. Instead, the assessment is made through manual resistance provided by the clinician. Although MMT is a quick and easy strength-testing method for clinicians, it is ineffective in detecting small-to-moderate strength changes in athletes [20,21]. It is also particularly unsuitable for subjects with considerable strength with an MMT score of 4 or higher [22].

Therefore, handheld dynamometers (HHD) were introduced based on manual resistance to provide a more objective measure of isometric muscle strength. However, despite this device’s user-friendliness, portability, and cost-effectiveness, the HHD’s reliability ranges from moderate to excellent [23,24]. Moreover, researchers have shown that the usability of HDDs depends on the operator who acquires the measure [25], and the reliability varies considerably depending on the muscle action tested [26]. For these reasons, their use is questionable, mainly when a slight deficit or imbalance in knee muscle strength is targeted. Due to such a lack of reliability, a few studies have aimed to find an association between isometric strength indices, assessed with HHD, and injury occurrence [27,28,29]. More recently, dynamometer anchoring system (DAS) [30,31,32,33] portable devices (with a dynamometer fixed in a platform), were developed to measure the isometric strength of lower limbs. DASs are portable and low-cost devices, but they have shown increased reliability compared to HDD [30]. One of the first DASs developed was tested by Nadler et al. [34] on ten healthy participants, showing high reliability for assessing hip adductor and extensor strength. Although they subsequently established the device’s value as a clinician tool, it has proven to be an excellent device for testing athletes during preparticipation sports physicals, when many participants are tested in succession [35].

Sung et al. [32] designed a DAS to measure the strength of the knee extensor in a supine position with the knee flexed at 35°. In this position, the DAS produced excellent intra- and interrater reliability for maximal isometric knee extensor strength measurements, with an interclass correlation coefficient (ICC) of 0.98 for both ratings and a 95% confidence interval (CI) of respectively 0.96–0.98 for intrareliability and 0.96–0.99 for interreliability. These results are similar to the findings of other HHD fixation studies for measuring the strength of knee extensors [36].

Ramsom et al. [30] demonstrated high reliability when testing maximal voluntary isometric knee flexion, hip abduction, and hip adduction with a fixed-frame dynamometry system. In particular, they evaluated the strength of knee flexors in a prone position in 30° and 45° hip and knee flexion, respectively, and with a foam roller under the pelvis. The system detected an ICC of 0.927 and 0.923 related to the maximum strength parameters of the left and right knee flexors.

In this paper, we describe a novel custom device to measure the isometric strength of knee extensors and flexors with the subject in a standing position. The device utilizes a portable frame and detachable load cells to test muscle strength. At the same time, dedicated software allows for the visualization of the signal acquired during the test and calculates the relative strength parameters.

This study aims to find the best setting of hip and knee inclination to measure the maximum voluntary isometric contraction (MVIC) of knee flexors and extensors and evaluate the device’s reliability with a test–retest procedure on male soccer players.

## 2. Materials and Methods

A custom device, shown in Figure 1a–g, was developed to measure the maximal contraction force of knee flexors and extensors in isometric conditions, keeping the subject in standing position.

The system components are shown in Figure 1a. A rigid exoskeleton was mounted on sliding guides that, together with adjustable U-shaped trunk support, allowed for the adjustment of the system to the subject’s anatomical parameters (height and leg length). In addition, a series of bands allowed for the stabilization of the trunk, the thigh, and the leg. The system had four load cells (Universal model CVP, Tecnologia & Soluzioni, Milano, Italy). Maximum load: 2 kN; repeatability: ±0.02%; (see Appendix A: CVP model load cell datasheet, containing all technical specifications) able to record both compression and traction forces. Two of them were positioned at the leg level and allowed measuring the contraction force of the knee extensor/flexor muscles (Figure 1b).

An acquisition board was connected to the four load cells and received/elaborated the force signals. First, a rigid link attached to the thigh could be shortened or elongated, thus perfectly adapting to the femur length. Then, it could be fixed through two handles (Figure 1d). A third handle allowed for the adjustment of the distance of the load cell, thus allowing it to put it in contact with the thigh (similarly, the leg load cell could be regulated to be in contact with the leg). Finally, four goniometers (Figure 1e) allowed for the verification of the flexion/extension degree of the hip and knee, which could be appropriately regulated.

The force testing procedure was carried out as follows: The subject stood without support and avoided holding with the arms. The hip and knee joint angles were set through the goniometers (see below for the exact values in each test). Figure 1f shows a subject performing a knee extensor test. Figure 1g shows the subject performing a knee flexor test (see Appendix A: Video showing the athlete’s preparation for an extensor force measurement).

Before starting the isometric force assessment test, the subject was instructed on the trial modality and carried out a mock test to become confident with the instrument: this consisted of a task in which the subject was asked to exert a muscle contraction at ~50% of the maximal one, followed by a maximal one. Then, the actual test started: the subject was asked to perform a rapid knee isometric contraction in extension (to evaluate the extensor force) or flexion (to evaluate the flexor force). The subject was also asked to keep the maximal muscle contraction force for 5 s before relaxing (see Appendix A: Video showing an isometric test to evaluate the maximal contraction of knee extensors).

For each test, three different measurements were carried out. First, a rest period of 30 s between the repetitions was used to avoid muscle fatigue affecting the measurement, similar to other state-of-the-art protocols [30,31]. Then, the measurement showing the highest force was chosen among the three and considered for the analysis. In the case of tests repeated on different days/weeks, they were always carried out at the same hour.

First, the optimal settings of the device were investigated in terms of the most suitable joint angles to be set for the subject undergoing the test. The repeatability of the measurement was assessed with a test–retest procedure carried out on male soccer players (Figure 2b).

### 2.1. Participants

As mentioned above, the study was divided into two phases: (1) procedure for finding the optimal angles and (2) test–retest analysis. For each phase, different subjects were recruited, and demographic and anthropometric data were collected at the beginning of the study (Table 1).

The first step involved eight healthy male subjects who were free of any medical concerns and practiced a regular (nonprofessional) sport activity at least twice per week. They had not suffered from knee or ankle injuries in the last 12 months and did not feel pain during maximal contraction of the knee extensor and flexor muscles.

The second phase involved 18 male subjects who played soccer in a nonprofessional team in Italy (demographic and anthropometric data are shown in Table 1). All players had completed one month of preseason training and were free from any lower limb injury. They were also free of any medical conditions. They had not suffered from knee or ankle injuries in the last 12 months and did not feel pain during maximal contraction of the knee extensor and flexor muscles.

Subjects were excluded from the study if they felt pain during the preliminary mock test. Moreover, athletes presenting a scar even near the support of the load cell were also excluded. None of the participants were excluded in the two phases of the study.

The investigation was conducted following the Declaration of Helsinki. As a result, institutional ethical approval was granted (ethics committee of University of Calabria; approval number 0076328, 19 October 2022). Before participating in the study, all subjects were informed of the benefits and risks, and informed written consent was subsequently obtained.

### 2.2. Procedures

#### 2.2.1. Procedure for Finding the Optimal Joint Angles

The first test aimed to find the optimal angles for the hip and the knee to be set as initial conditions for the subsequent experiments. This analysis was conducted in one day through a series of measurements at different angles on a group of 8 subjects (Figure 2a). This first step allowed for the identification of the optimal starting machine settings for quantifying at best muscle strength of knee extensors and flexors in a standing position.

While subjects underwent maximal force recordings for knee extensors and flexors using the machine, at the same time, surface electromyographic measurements (FREEEMG, BTS Bioengineering) at the *rectus femoris*, *vastus medialis*, long head of the *biceps femoris*, and *semitendinosus* were performed. After appropriate skin cleaning, two electrodes were attached by an expert operator 0.02 m apart (center-to-center) on the skin, above each muscle, halfway between the center of the belly and the distal myotendinous junction.

These measurements were made by varying the hip and knee initial inclination. In the literature, many devices set the hip at a fixed flexion angle of 90°. In these conditions, it is known that the maximal extensor force can be obtained with the knee flexed between 60° and 70°, while the maximal flexor force can be obtained with the knee flexed at about 40° [36,37]. Thus, in our analysis, we first fixed the knee flexed to 70° and varied the hip’s flexion from −30 to +40°, recording the maximal extensor force. Then, we repeated the procedure with knee flexion fixed at 40°, recording the maximal flexor force. The results obtained allowed for the setting of the optimal hip angles, which resulted in +20° for the extensors and +30° for the flexors (see the results section) after establishing the optimal hip flexion angles. Finally, we reevaluated the strength and muscle activation profiles as the knee angle changed, namely 30, 40, and 50° for both flexors and extensors, in order to find the knee angles that produced the maximum strength of the two muscle groups for these specific angles of the hip. Each test was performed two times, in which we acquired MVIC value recorded by the exoskeleton and maximum contraction value (MCV) extracted by EMG signal at a sampling rate of 1 kHz.

#### 2.2.2. Test–Retest Analysis

In the test–retest analysis, 18 subjects were asked to perform the same task with the machine three times in the space of one week. This procedure allowed us to quantify the reliability of the measurements. The subjects performed a mock test a few weeks before the test–retest analysis to familiarize themselves with the procedure and minimize the possible learning effect. The test–retest sessions were carried out in the middle of the week (as far as possible from training and matches), the day after a resting day, by the same operator in a controlled environment.

The knee extensor and flexor MVIC force values were assessed on the 18 subjects at the first time-point (TR1), after one week (TR2), and after two weeks (TR3). As mentioned, the same procedure and time of the day were maintained for the three time-points. During those two weeks, the subjects were not subjected to specific force training programs but only to regular soccer training (twice a week).

### 2.3. Analysis

In the test–retest reliability analysis, two variables were evaluated: absolute repeatability and relative repeatability.

Typical error % (*TE*%) was calculated as follows:(1)$$TE\%=\left(SD\left(TR_{x}-TR_{y}\right)\sqrt{2}\right)\times 100$$

*SD* was the standard deviation of the differences between two measurements (*TR_x_* and *TR_y_*). ICC was the intraclass correlation coefficient calculated with the Hopkins model [36,37]. Both indexes were expressed with the corresponding 90% confidence interval for each parameter under analysis, namely the maximal force of the right knee extensor, the left knee extensor, the right knee flexor, and the left knee flexor.

All analyses were performed using SPSS 26.0 software (SPSS, Chicago, IL, USA) running on Windows. The significance level was set in all mentioned analyses at a *p*-value < 0.05.

## 3. Results

### 3.1. Optimal Joint Angle Analysis

The results of the hip and knee optimal angle analyses are reported in Figure 3a–d.

We considered the optimal angles corresponding to the maximum muscle contraction force registered by the device in Figure 1 and confirmed by the EMG signals. Overall, these results allowed for the setting of the optimal angles for using the instrument, namely the hip flexing at 20° and knee flexing at 80° when measuring the force of extensor muscles, and the hip flexing at 30° and knee flexing at 40° when measuring the force of the flexor muscles.

### 3.2. Test–Retest Reliability

The results obtained in terms of test–retest reliability are reported in Table 2.

Overall, these results demonstrate the excellent reliability of the instrument over the tested period.

## 4. Discussion

The device for isometric force measurement proposed in this study was first evaluated in terms of the optimal angles set at the hip and the knee to achieve the maximal contraction force performance. Many devices described in the literature have a fixed hip angle (90°). It has been highlighted that in these conditions, flexing the knee between 60° and 70° allows for a maximal extensor force, while flexing the knee at 40° allows for the obtaining of a maximal flexor force [36,37,38]. Concerning the state of the art, we identified different angles as optimal due to the device’s different architecture and working principles. The results obtained by monitoring the maximal contraction force registered by the device were supported by EMG signals, which have been demonstrated to be reliable and repeatable for assessing force contraction performance [39]. The optimal angles for using the instrument were 20° of flexion for the hip and 80° for the knee (Figure 3) when measuring the force of knee extensor muscles, and 30° of flexion for the hip and 40° of flexion for the knee when measuring the force of knee flexor muscles. This agrees with the joint inclinations of Ransom et al. that were recently used for the flexor test in a prone position using an anchored dynamometry system [30].

The results demonstrated high relative reliability when testing maximal voluntary isometric knee flexion and extension using a portable dynamometer anchoring system (DAS). This study is the first to investigate a device’s reliability for measuring the standing position of knee flexors and extensor strength in soccer players.

As shown in Table 2, ICC values ranged from 0.96 to 0.98 for knee extensors and from 0.91 to 0.97 for knee flexors. The ICC values were higher than 0.9 and much higher than 0.95. These data demonstrate the excellent reliability of the instrument. ICCs > 0.75 represent good reliability, 0.50 to 0.75 moderate reliability, and <0.50 poor reliability [39]. Our findings were in line with the ones reported in recent studies: Desmyttere and colleagues claimed as “good to excellent” the reliability of a hip strength assessment system whose ICC values resulted in the range of 0.77–0.95 [40]. Sung et al. found ICC values of 0.96–0.98 for a portable dynamometer anchoring system to be used in a supine position to assess knee extensor strength [32]. Mentiplay and colleagues claimed a good-to-excellent test–retest reliability (ICCs = 0.82–0.97) for a handheld dynamometer for isometric strength measurements by performing measurements at two different time-points [41]. Similar results were shown by Hogrel, reporting an ICC value of 0.98 for a novel portable device used to measure the isometric force of the knee extensors [42].

The advantages of this test are lower cost if compared to the gold standard, and the method being easy to manage even for an inexperienced operator. In addition, the device is portable, and the measurements are recorded rapidly, so the procedure could be part of a routine athlete monitoring or screening program.

The contraction capacity of a muscle and its maximum strength are also linked to its state of lengthening/shortening. A muscle can express a greater force in a specific intermediate position between the maximum stretch and the maximum shortening. Seated strength assessment forces the hip into a position of approximately 90 degrees of flexion. In this position, the biarticular muscles of the knee’s extensor compartment and the flexor compartment are in a nonoptimal position for the expression of maximum force (respectively in excessive shortening and excessive lengthening) [43,44]. On the other hand, with the support of electromyographic feedback, in the erected station, we varied the angles of both the knee and the hip. Hence, it allowed us to evaluate which exact position, knee flexion, and hip flexion express the most significant recruitment of muscle fibers, and therefore, the maximum strength.

One possible limitation of the study regards the optimal joint angle analysis, in which the sample size investigated was small but still sufficient to draw significant results. In addition, also at this stage of this research, we limited the analysis to the study of specific hip and knee angles based on literature values and physiological considerations to minimize the effect of fatigue that would have occurred if the subject had performed too many tests [45,46]. Nevertheless, we cannot claim to have excluded the effect of fatigue in the procedure for searching for optimal angles. Finally, the test–retest analysis involved male soccer players; extrapolation of results to other sporting and nonsporting populations should, therefore, be considered carefully.

Future research should assess the device for measuring knee flexor and extensor strength on a larger sample size and examine the associations between the strength-based parameters with other dynamometry assessments that have to provide sound in injury screening. Furthermore, the device should screen flexor and extensor strength longitudinally to determine if this test could predict injuries.

## 5. Conclusions

This study presents a custom device that is useful for isometric strength assessments on athletes. First, optimal joint angles for the hip and the knee were found and set as the test’s initial conditions. Then, the reliability of the measurements enabled by the device was assessed, finding excellent results over the two weeks of the testing period. The DAS allows for repeatable assessment of muscle strength of knee flexors and extensors, potentially allowing for efficient detection of changes in muscle strength that may provide information as to the athlete’s musculoskeletal status. Although this could be the first step, the device needs further studies, perhaps in comparison with other methods and evaluating whether it is possible to correlate the outcomes with practice, for example, training.

## Figures and Tables

**Figure 1 life-13-00458-f001:**
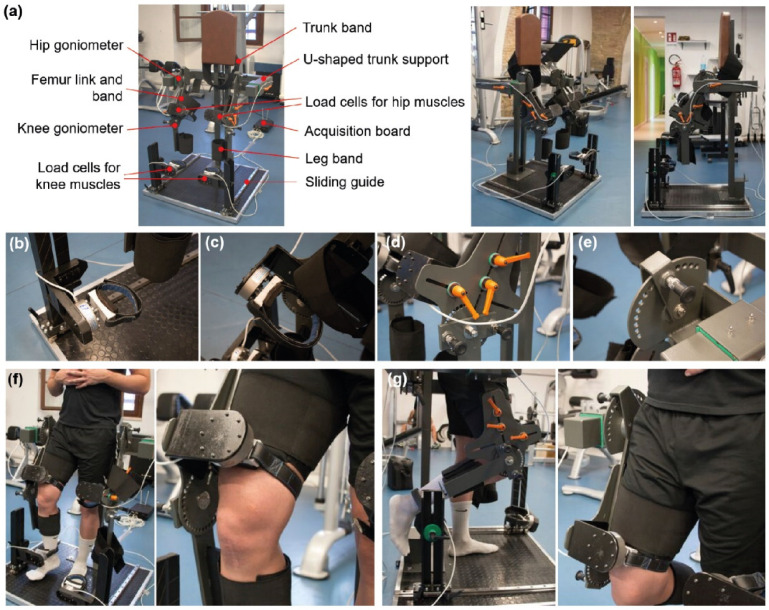
Description of the custom device to measure the isometric force used in this study. Scheme of the instrument with the critical components highlighted and pictures of the device from different viewpoints (**a**). Zoomed views highlight some essential components of the platform: load cell at the ankle (**b**); load cells at the knee (**c**); handles allowing one to adapt the machine dimensions to the subject anatomical characteristics (**d**); and goniometer to verify the flexion/extension degree of the joint (**e**). (**f**) Images of a subject undergoing a knee extensor force measurement test; (**g**) images of a subject undergoing a knee flexor force measurement test.

**Figure 2 life-13-00458-f002:**
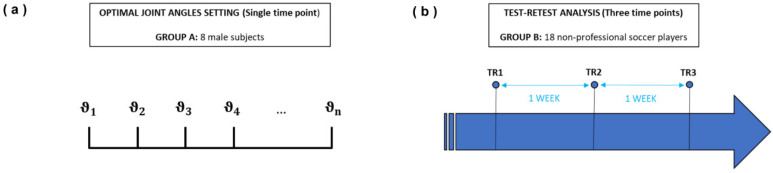
Design of the study, divided into two phases: optimal angle analysis for finding the best initial device settings (**a**); and test–retest analysis to verify the reliability of the device measurements and force conditioning (**b**).

**Figure 3 life-13-00458-f003:**
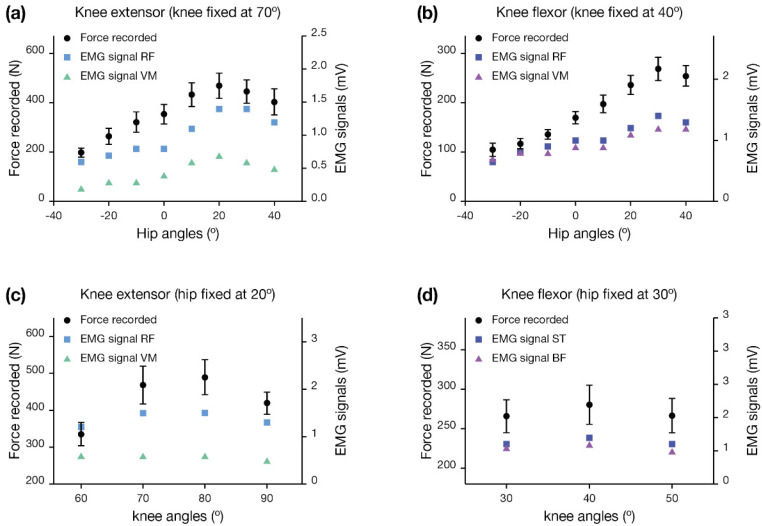
Identification of optimal hip and knee joint angles for machine setting. Force (left Y axes) and average values of EMG signals (right Y axes) are shown for different conditions: (**a**) knee extensor isometric contractions, with knee fixed at 70° and varying hip angles; (**b**) knee flexor isometric contractions, with knee fixed at 40° and varying hip angles; (**c**) knee extensor isometric contraction, with hip fixed at 20° and varying knee angles; (**d**) knee flexor isometric contractions, with hip fixed at 30° and varying knee angles. R.F. = rectus femoris; V.M. = vastus medialis; S.T. = semitendinosus; B.F. = biceps femoris.

**Table 1 life-13-00458-t001:** Demographic and anthropometric data.

Descriptive Measures	Optimal Joint Angles Setting (*n* = 8)	Test–Retest Analysis (*n* = 18)
Age (years)	29.75 ± 3.85	24.32 ± 3.23
Height (cm)	182.0 ± 5.8	178.9 ± 6.9
Weight (kg)	74.4 ± 6.9	73.3 ± 7.6

**Table 2 life-13-00458-t002:** Results of the test–retest analysis.

		Right Knee Extensor (Max Contraction)	Left Knee Extensor (Max Contraction)	Right Knee Flexor (Max Contraction)	Left Knee Flexor (Max Contraction)
TR1-TR2	**TE%**	**2.90**	**3.40**	**3.60**	**4.30**
	Lower; Upper CI	(2.20; 4.00)	(2.60; 4.70)	(2.80; 5.10)	(3.30; 6.00)
	**ICC**	**0.98**	**0.97**	**0.96**	**0.96**
	Lower; Upper CI	(0.95; 0.99)	(0.93; 0.98)	(0.91; 0.98)	(0.90; 0.98)
	**Change in the Mean %**	**1.70**	**1.06**	**1.42**	**1.66**
	Lower; Upper CI	(0.03; 3.38)	(−0.87; 3.03)	(−0.65; 3.53)	(−0.77; 4.15)
TR2-TR3	**TE%**	**2.50**	**3.40**	**3.80**	**3.20**
	Lower; Upper CI	(1.90; 3.50)	(2.60; 4.80)	(2.90; 5.30)	(2.40; 4.40)
	**ICC**	**0.98**	**0.97**	**0.96**	**0.97**
	Lower; Upper CI	(0.95; 0.99)	(0.92; 0.98)	(0.91; 0.98)	(0.93; 0.98)
	**Change in the Mean %**	**−0.84**	**0.11**	**6.21**	**3.79**
	Lower; Upper CI	(−2.26; 0.60)	(−1.82; 2.07)	(3.95; 8.51)	(1.92; 5.68)
TR1-TR3	**TE%**	**3.70**	**4.00**	**5.50**	**4.10**
	Lower; Upper CI	(2.80; 5.10)	(3.10; 5.60)	(4.20; 7.70)	(3.10; 5.70)
	**ICC**	**0.96**	**0.96**	**0.91**	**0.96**
	Lower; Upper CI	(0.91; 0.98)	(0.90; 0.98)	(0.81; 0.96)	(0.90; 0.98)
	**Change in the Mean %**	**0.84**	**1.17**	**7.72**	**5.51**
	Lower; Upper CI	(−1.23; 2.96)	(−1.08; 3.48)	(4.45; 11.09)	(3.09; 7.97)

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
