# Peer review of "Reliability of a Custom Device Used to Measure Isometric Knee Flexor and Extensor Strength in Standing Position"

_life, 2023, doi:10.3390/life13020458_

Round 1

Reviewer 1 Report

I invite you to resolve the situation found in the ranks129, 143, 152, 156, 158, 159, 163, 164, 182, 184, 218, 225, 252

Please add date of the approval number (0076328)

Author Response

We thank the reviewer for the useful suggestions.

I invite you to resolve the situation found in the ranks129, 143, 152, 156, 158, 159, 163, 164, 182, 184, 218, 225, 252

We fixed all.

Please add date of the approval number (0076328)

We fixed all.

Reviewer 2 Report

Dear Authors, 

Your paper is out of the scope of the special issue: Sports Medicine: Nutritional Sciences and Nutritional Biochemistry

Author Response

We agreed, and the manuscript was switched.

Reviewer 3 Report

First, congratulations to the authors for their effort and interest in this relevant topic. The following are my observations.

 Abstract.

Conclusion. It should be based on the findings of reliability, not validity.

 1.- Introduction.

- The acronym MVIC must be specified

- Please provide more clarity to establish the knowledge gap or problem.

 2.- Materials and Methods

- Correct the errors : “Error! Reference source not found”

- The table names must be at the top.

- Table 1 is repeated and not referenced in the text.

- What is TE%? Please define.

  3.- Results

- The table names must be at the top.

 4.- Discussion.

- Please argue to a greater extent the contribution of the evaluation in the bipedal position, with respect to the evaluation in the seated position.

Author Response

First, congratulations to the authors for their effort and interest in this relevant topic. The following are my observations.

We thank the reviewer for the appreciation

 Abstract.

Conclusion. It should be based on the findings of reliability, not validity.

Thank for the suggestion we added a line in the abstract

 1.- Introduction.

The acronym MVIC must be specified

We added the full meaning

- Please provide more clarity to establish the knowledge gap or problem.

We tried to elucidate better

 2.- Materials and Methods

Correct the errors : “Error! Reference source not found”

We fixed all

The table names must be at the top.

We corrected

Table 1 is repeated and not referenced in the text.

We do not know why the PDF was doubled, anyway, we corrected and cited in the text

What is TE%? Please define.

We defined

  3.- Results

The table names must be at the top.

We switched

 4.- Discussion.

Please argue to a greater extent the contribution of the evaluation in the bipedal position, with respect to the evaluation in the seated position.

We added a better explanation 

We hope that with those improvements the manuscript should be suitable for publication

Round 2

Reviewer 2 Report

Dear authors,

The paper is out of the scope of the SI.